Expression and correlation of COX-2 and NUCB1 in colorectal adenocarcinoma

Zhang Yuanyuan 1
Zhang Gai 2
Zhong Jinghua 1
Li An 1
Wu Yanyang 1
Guo Zhenli 1 wycg1984@163.com
1 Department of Oncology, First Affiliated Hospital of Gannan Medical University , Ganzhou , China
2 Department of Emergency Internal Medicine, The First Affiliated Hospital of Wannan Medical College Yijishan Hospital , Wuhu , China
Dong Peixin
Electronic publication date: 2023 Jul 31
Publication date: 2023
Volume: 11
Electronic Location ID: e15774
Received 2023 May 29; Accepted 2023 Jun 29
Copyright: © 2023 Zhang et al.
Copyright year: 2023
Copyright holder: Zhang et al.
License: This is an open access article distributed under the terms of the Creative Commons Attribution License, which permits unrestricted use, distribution, reproduction and adaptation in any medium and for any purpose provided that it is properly attributed. For attribution, the original author(s), title, publication source (PeerJ) and either DOI or URL of the article must be cited.
License URL: https://creativecommons.org/licenses/by/4.0/

Keywords: Colorectal adenocarcinoma, Cyclooxygenase-2, Nucleobindin-1, Immunohistochemistry

Funding: Science and Technology Program of Jiangxi Provincial Administration of Traditional Chinese Medicine 2022A035 Wuhu Science and Technology Plan Project 2021yf67 This project was supported by the Science and Technology Program of Jiangxi Provincial Administration of Traditional Chinese Medicine (2022A035) and by the Wuhu Science and Technology Plan Project (2021yf67). The funders had no role in study design, data collection and analysis, decision to publish, or preparation of the manuscript.

==============================
Objective

To investigate the expression and correlation of COX-2 and NUCB1 in colorectal adenocarcinoma and adjacent tissues.

Methods

The expression of COX-2 and NUCB1 and their effects on prognosis were predicted using bioinformatics. Immunohistochemistry was used to identify the expression of two molecules in 56 cases of colorectal adenocarcinoma and the surrounding tissues. The expression of two molecules and their association with clinicopathological variables were examined using the chi-square test. The association between COX-2 and NUCB1 was investigated using the Spearman correlation test.

Results

The STRING database revealed that COX-2 and NUCB1 were strongly linked. According to the UALCAN and HPA database, COX-2 was upregulated while NUCB1 was downregulated in colorectal adenocarcinoma, both at the protein and gene levels. The OS times for COX-2 and NUCB1 high expression, however, exhibited the same patterns. The rate of positive COX-2 immunohistochemical staining in cancer tissues was 69.64% (39/56), which was significantly higher than the rate in healthy tissues 28.57% (16/56). NUCB1 was expressed positively in cancer tissues at a rate of 64.29% (36/56) compared to just 19.64% (11/56) in neighboring tissues. The positive expression levels of COX-2 and NUCB1 were both closely related to clinical stage, differentiation degree, and lymphatic metastases (P < 0.05). In colorectal cancer, COX-2 and NUCB1 expression were significantly correlated (rs = 0.6312, P < 0.001).

Conclusion

Both COX-2 and NUCB1 are overexpressed and significantly associated in colorectal adenocarcinoma.

Introduction

Bevacizumab is commonly applied to patients with middle-to-advanced colorectal adenocarcinoma with metastasis. By binding to vascular endothelial growth factor-A (VEGF-A), it blocks VEGF-A-induced angiogenesis, consequently inhibiting tumor metastasis and prolonging patients’ survival (Midgley & Kerr, 2005). However, bevacizumab is prone to drug resistance, which brings great difficulties to cancer treatment (Midgley & Kerr, 2005; Hurwitz et al., 2004). Many studies have shown that interfering with the expression of VEGF can affect the efficacy of bevacizumab (Feng et al., 2020). COX-2 is one of the influencing factors. Our preliminary clinical data once again confirmed the correlation between COX-2 and bevacizumab resistance. COX-2-induced VEGF overexpression may be a mechanism of bevacizumab resistance (Rao et al., 2004). Nucleobindin1 (NUCB1) was first identified in systemic lupus erythematosus (SLE) and was characterized by a variety of functional binding domains, including DNA binding domains, nuclear localization signals, etc., which gave it many functions (Sinha, Pattnaik & Aradhyam, 2019). Structurally, the supposed binding domain of NUCB1 can bind with COX-2, and they can be colocalized (Leclerc et al., 2008). Functionally, the addition of hrNUCB to purified hrCOX-2 showed increased PGE2 synthesis, suggesting that NUCB enhances COX-2 activity (Leclerc et al., 2008). Therefore, it is reasonable to speculate that NUCB1 and COX-2 are closely related. Immunohistochemistry was applied to detect the expression of COX-2 and NUCB1 in colorectal adenocarcinoma and adjacent tissues. By analyzing their relationship with clinicopathologic parameters, as well as their correlation, it is possible to explore a potential pathway for the occurrence and development of colorectal cancer. It may pave the way for exploring the mechanism of COX-2-induced bevacizumab resistance.

Materials and Methods

Patients

This retrospective study was carried out for the case series of our hospital. We selected 10% of the formalin-fixed, paraffin-embedded colorectal cancer specimens from the pathology department of the Gannan Medical University’s First Affiliated Hospital between June 2019 and June 2020. Eligible patients were screened according to the following criteria: the pathological diagnosis was confirmed as colorectal adenocarcinoma; radical resection of colorectal cancer; mucinous adenocarcinoma was excluded; preoperative treatments, including neoadjuvant chemotherapy, targeted therapy, and traditional Chinese medicine, were excluded. Metastatic and recurrent colorectal cancer were excluded. Cases with two or more types of tumors combined with cardiovascular disease and other major diseases were excluded. Totally, 56 cases have been screened, and each case has been sliced into three slices of the cancer tissue and the adjacent tissue to perform immunohistochemistry staining. The classification and count of clinicopathological parameters are shown in Table 1.

Table 1 Classification and counting of clinicopathologic parameters of colorectal adenocarcinom.

Clinicopathologic parameters	n	Percent (%)	
Gender			
Male	35	62.5	
Female	21	37.5	
Age (years)			
≥50	51	91.07	
<50	5	8.93	
Site			
Left colon	38	67.86	
Right colon	18	32.14	
Differentiation			
High differentiation	4	7.14	
Middle differentiation	40	71.43	
Low differentiation	12	21.43	
TNM stage			
I	11	19.64	
II	17	30.36	
III	24	42.86	
IV	4	7.14	
Serosa infiltration			
Yes	44	78.57	
No	12	21.43	
Lymphatic metastases			
Yes	35	62.50	
No	21	37.50	
Distant metastases			
Yes	4	7.14	
No	52	92.86	

The study has been conducted in accordance with the Declaration of Helsinki and was approved by the First Affiliated Hospital of Gannan Medical University’s Ethics Committee (approval number ZZSL 2023-104). Due to the retrospective nature of this study, patient informed consent was waived.

Methods

Bioinformatics analysis

The UALCAN database can forecast pan-oncogene expression and survival information, and evaluate gene promoter methylation (Chandrashekar et al., 2022, 2017).

The STRING database is one of the most abundant and widely used databases that can predict protein interactions at present (Szklarczyk et al., 2023). It can not only generate the interaction diagram, but also demonstrate the correlation analysis of proteins, including GO analysis, KEGG analysis, etc.

The Human Protein Atlas (HPA) is based on proteomics, transcriptomics, and systems biology data, which can map tissues, cells, organs, etc. Not only tumor tissue but also the protein expression of normal tissues is covered (Karlsson et al., 2021).

The Kaplan–Meier plotter is a website focused on online survival analysis (Győrffy, 2023).

Immunohistochemistry

A concentrated murine anti-human NUCB1 monoclonal antibody was purchased from ORIGENE, and a ready-to-use rabbit anti-human COX-2 monoclonal antibody and instant immunohistochemical MaxVision™ kit were purchased from Fuzhou Maixin Biotech Ltd., Fujian, China. The paraffin tissue was dewaxed and hydrated, then heated in citrate buffered saline to repair antigen and soaked in a 3% hydrogen peroxide solution to block endogenous peroxidase. The diluted concentration of NUCB1 was 1:150. PBS was used as a negative control instead of the primary antibody, and the reagent buyer provided a positive control. IHC used a two-step approach, the SP method. We referred to the kit instructions for specific steps.

Immunohistochemical evaluation

The results of immunohistochemical staining were evaluated by two senior pathologists. The cytoplasm or nucleus with varying shades of yellow was regarded as positive. Five consecutive visual fields (400×) were selected for positive cell count and staining intensity judgment. The results were estimated by semi-quantitative integration (Guo et al., 2011) (Table 2).

Table 2 Semi-quantitative integration.

Positive cell (%)	Staining intensity	Multiply	
Negative	0	Negative	0	0–4	–	
≤10	1	Pale yellow	1	4–6	+	
11–50	2	Brownish yellow	2	6–9	++	
51–75	3	Brown	3	9–12	+++	
>75	4					

Statistical analysis

GraphPad Prism 8.0 was used to conduct each statistical analysis. A chi-square test was used to determine whether COX-2 and NUCB1 expressions were statistically significant in colorectal adenocarcinoma and whether they were associated with things like differentiation, TNM staging, location and lymphatic metastases, and other things. The Spearman correlation test was used to analyze the correlation between COX-2 and NUCB1. P < 0.05 was considered to have a statistical difference.

Results

Bioinformatics analysis

The UALCAN database showed that the expression of COX-2 is up-regulated in cancer tissues, especially in stages II and III (Figs. 1A and 1B, P < 0.01), suggesting that COX-2 may be a cancer-promoting factor in colorectal adenocarcinoma. The STRING database provided clues to the correlated molecules of COX-2. Figure 2A only showed a part of them. After preliminary screening, six molecules were identified, namely APC, CAV-1, CTNNB1, MMP7, NUCB1, and DUSP1. Owing to the fact that the correlation between NUCB1 and COX-2 in colorectal adenocarcinoma has not been reported, we finally locked on NUCB1 (Fig. 2B predicts the correlation between COX-2 and NUCB1). The UALCAN database showed that the expression of NUCB1 was decreased in cancer tissues of stages I to IV, suggesting that NUCB1 was a negative regulatory molecule, probably (Figs. 1C and 1D, P < 0.01).

Figure 1 COX-2 was significantly increased in colorectal adenocarcinoma (A). COX-2 expression was increased in stages II and III (B). NUCB1 was significantly decreased in colorectal adenocarcinoma (C). NUCB1 expression decreased in stages I to IV (D).

An asterisk (∗) indicates that the results are statistically significant.

Figure 2 The STRING database predicted s molecules associated with COX-2 (A). NUCB1 i was related to COX-2 (B).

RFS, OS, and PPS times were predicted by the Kaplan–Meier Plotter database (Fig. 3). In comparison to low expression, COX-2 high expression had a longer RFS time (P < 0.05). NUCB1 was the contrary (P < 0.05), though. OS time of COX-2 and NUCB1 had the same trends (P < 0.05). The PPS time of COX-2 high expression was shorter than that of low expression (P < 0.05). However, the PPS time of NUCB1 was not statistically significant.

Figure 3 RFS, OS, and PPS time between high and low expression of COX-2 and NUCB1 in COAD from the Kaplan–Meier plotter.

The red and black lines represent high and low expression, respectively.

The expression of COX-2 and NUCB1 in colorectal cancer was predicted using the Human Protein Atlas (HPA) (Fig. 4). In healthy colon tissue, NUCB1 was highly expressed while COX-2 was not present. COX-2 and NUCB1 expression ranged from low to high in cancerous tissue.

Figure 4 Expressions of COX-2 and NUCB1 in normal colon tissue and colorectal adenocarcinoma tissue from the Human Protein Atlas (HPA).

(A–D) referred to COX-2 and (E–H) referred to NUCB1.

The expression of COX-2 and NUCB1 in immunohistochemistry

COX-2 and NUCB1 were mainly located in the cytoplasm. In 56 cases of cancer tissues, 39 showed different levels of COX-2, with a positive rate of 69.64%, whereas only 16 cases and 28.57% were in the adjacent cancer tissues correspondingly. The chi-square test is X2 = 19.85 as well as P < 0.005, confirming that COX-2 was significantly upregulated in colorectal adenocarcinoma. There were 36 cases with a positive NUCB1, accompanied by a positive expression rate of 64.29%, corresponding to 11 cases and 19.64% in the cancerous adjacent tissues. The chi-square test result was X2 = 24.79, with a P < 0.005, indicating that NUCB1 was markedly increased in colorectal adenocarcinoma (As shown in Fig. 5 and Table 3).

Figure 5 (A–E) were COX-2, and (F–J) were NUCB1 through IHC (400×).

COX-2 and NUCB1 were negative in adjacent tissues (A, F), SP (400×). Negative expression of COX-2 and NUCB1 in cancer tissues (B, G), SP (400×). In cancer tissue, COX-2 and NUC.

Table 3 Expressions of COX-2 and NUCB1 in colorectal adenocarcinoma.

	n	COX-2	Positive rate	P	NUCB1	Positive rate	P	
–	+	++	+++	–	+	++	+++	
Cancer tissue	56	17	10	12	17	69.64%	P < 0.005	20	17	8	11	64.29%	P < 0.005	
Adjacent tissue	56	40	6	3	7	28.57%	45	8	2	1	19.64%	

Relationship of COX-2 and NUCB1 with clinicopathological parameters

As shown in Table 4, COX-2 was significantly relevant to differentiation degree (X2 = 8.18, P < 0.05), clinical stage (X2 = 9.80, P < 0.05), and lymphatic metastases (X2 = 4.72, P < 0.05). The higher the clinical stage and differentiation degree, the higher the positive rate of COX-2, which was 100% in highly differentiated tissues. No correlation was manifested between COX-2 and gender (X2 = 2.48, P > 0.10), age (X2 = 0.28, P > 0.50), location (X2 = 0.11, P > 0.50), infiltration of serosa (X2 = 1.35, P > 0.10), distant metastases (X2 = 1.87, P > 0.10). Whereas COX-2 was positively expressed in all cases with distant metastases.

Table 4 Relationship of COX-2 and NUCB1 with clinicopathological parameters.

Clinicopathological parameters	COX-2	NUCB1	
	Positive rate (%)	P	Positive rate (%)	P	
Gender		P > 0.10		P > 0.10	
Male	77.14		71.43		
Female	57.14		52.38		
Age (years)		P > 0.50		P > 0.50	
≥50	68.63		64.71		
<50	80.00		60.00		
Site		P > 0.50		P > 0.10	
Left colon	71.05		57.89		
Right colon	66.67		77.78		
Differentiation		P < 0.05		P < 0.05	
High-differentiation	100.00		75.00		
Mid-differentiation	72.50		70.00		
Low-differentiation	50.00		41.67		
TNM stage		P < 0.05		P < 0.05	
I	54.55		45.45		
II	58.82		58.82		
III	79.17		62.96		
IV	100.00		100.00		
Serosa infiltration		P > 0.10		P > 0.10	
Yes	65.91		63.64		
No	83.33		66.67		
Lymphatic metastases		P < 0.05		P < 0.05	
Yes	80.00		74.29		
No	52.38		47.62		
Distant metastases		P > 0.10		P > 0.50	
Yes	100.00		100.00		
No	67.31		61.54		

NUCB1 and differentiation degree (X2 = 6.89, P < 0.05), clinical stage (X2 = 9.18, P < 0.05), and lymphatic metastases (X2 = 4.07, P < 0.05) existed statistically significant, but not with gender (X2 = 2.07, P > 0.10), age (X2 = 0.04, P > 0.50), tumor site (X2 = 2.11, P > 0.10) and invasion of serosa (X2 = 2.11, P > 0.10), distant metastases (X2 = 0.22, P > 0.50). Patients with middle-to-advanced stages or well-differentiated tumors had a higher positive expression rate of NUCB1.

The correlation between COX-2 and NUCB1

The Spearman correlation test showed that the correlation coefficient between COX-2 and NUCB1 was rs = 0.6312, with P < 0.001, which indicated that a remarkably strong positive correlation should be initially recognized.

Discussion

COX-2 is nearly not present in normal tissues but can be induced by growth factors, cytokines, TNF-α, and vascular activator peptides to participate in tumor growth, metastasis, invasion, and angiogenesis (Wang & Dubois, 2006; Zhan et al., 2004). Through bioinformatics analysis, we discovered that COX-2 in colorectal cancer was significantly up-regulated, especially in stages II and III. And COX-2 high expression had the highest probability in terms of RFS time but opposite in terms of OS and PPS time. COX-2 was negative in normal colon tissue and presented to varying degrees as positive in cancer tissue. These results suggest that COX-2 may be a potential promoter factor. According to immunohistochemistry, the positive rate of COX-2 in cancer tissues was 69.64%, which was a lot higher than the negative rate of 28.57% in healthy tissues nearby (P < 0.05). It validates previous bioinformatics analysis and basically conforms to the erstwhile experimental results of approximately 85% of colorectal cancer expressing COX-2 and about 20% of normal colorectal tissues (Eberhart et al., 1994; Ogino et al., 2008; Zhang & Sun, 2002). In this research, COX-2 was closely correlated with TNM staging, differentiation degree, and lymphatic metastases. And the proportion of COX-2 expression increased with TNM staging. It gives an insight into the real possibility that COX-2 may provide advantageous conditions for the growth of cancer cells and aggravate their invasiveness and migration. Owing to the decreasing differentiation degree, the positive rate of COX-2 also decreased, indicating that COX-2 may be involved in the differentiation process of colorectal cancer. In TNM stage IV and distant metastases, the expression rate of COX-2 reached 100%. Despite the fact that there are so few advanced cases, it is possible to speculate that COX-2 is almost exclusively expressed in advanced, high-grade colorectal cancer. It was consistent with previous research results: COX-2 was associated with the advanced stage and high proliferation activity of colorectal cancer (Zhang & Sun, 2002). Therefore, COX-2 may be a contributing factor and a vital target for diagnosis and treatment. In addition, there was no statistical difference in COX-2 and age, gender, distant metastases, or serosal invasion, which may be attributed to an insufficient sample. We will continue to expand the sample and attempt to perfect the relevant results. Not only in colorectal cancer, but also in several animal models, selective COX-2 inhibitors can greatly weaken cancer growth. COX-2 may be a crucial factor in the occurrence and deterioration of cancer. Some evidence has been provided that exogenous induction of COX-2 in human oral epithelial keratinocytes can trigger the proliferation of the keratinocytes and secretion of cyclin D1, taking a further step toward malignant transformation (Wang et al., 2019). COX-2 is a key enzyme to generate PGE2, which can enhance the proliferation of ovarian cancer cells by binding to the corresponding receptor and then catalyzing the phosphorylation of NF-κB/P65 (Zhang et al., 2019). In light of this evidence, we will continue to detect COX-2 mRNA and protein expression at the cellular and mouse levels to verify the function of COX-2 in colorectal cancer and go a step further toward exploring whether COX-2 is an important cause of bevacizumab resistance.

Through the retrieval of the STRING database, we initially screened out six molecules, namely APC, CAV-1, CTNNB1, MMP7, NUCB1, and DUSP1, and then searched and analyzed them by PubMed. We found that there were few relevant studies to NUCB1, such as expression level and function. The relationship between NUCB1 and COX-2 has also only been mentioned in vitro. Accordingly, we chose NUCB1 as the object of study. NUCB1 was first discovered in systemic lupus erythematosus (SLE), which bound to nucleosome ladder DNA to induce autoimmunity and thymus apoptosis (Miura et al., 1992; Kanai et al., 1995). The most typical feature of NUCB1 is its multiple functional domains: DNA binding sites, heterodimerization domains, two EF motifs, Ca2+ binding sites, nuclear localization signals, leucine zipper regions, and non-classical domains, including G-protein binding domains and cyclooxygenase binding domains (Miura et al., 1992). These endow NUCB1 with important implications in a variety of cellular processes, including autoimmunity, intracellular signaling, osteogenesis, inflammation, and cancer. For example, the DNA binding domain of NUCB1 can combine with the E-box sequence of the Cripto promoter to activate many intracellular signaling pathways and trigger the biological behavior of cancer cells (Sinha, Pattnaik & Aradhyam, 2019). The non-classical binding domain of NUCB1 binds to cyclooxygenase, which has been confirmed by yeast two-hybrid systems and immunoprecipitation experiments (Ballif et al., 1996). This gives us confidence to continue our research. Through bioinformatics, NUCB1 in colorectal cancer tissues was reduced compared to normal tissue, both at the mRNA and protein levels. However, the OS time of NUCB1 high expression was shorter. These results are contradictory. In this study, compared to 19.64% in adjacent tissues, the positive rate of NUCB1 in cancer tissues was significantly increased to 64.29% (P < 0.05), which is consistent with the research results of Sinha, Pattnaik & Aradhyam (2019). In addition, NUCB1 was significantly relevant to TNM stage, differentiation degree, and lymphatic metastases, suggesting that NUCB1 may induce the proliferation and metastasis of colorectal cancer. The higher the differentiation degree of colorectal cancer, the higher the positive rate of NUCB1, indicating that NUCB1 may be involved in the differentiation of colorectal cancer. The Spearman correlation test showed that the correlation coefficient between NUCB1 and COX-2 is rs = 0.6312 and P < 0.001, revealing a significant positive correlation between NUCB1 and COX-2. This study did not show any statistical difference between NUCB1 and gender, age, tumor site, serosal invasion, or distant metastases. Interestingly, NUCB1, along with COX-2, was highly expressed in TNM stage IV and distant metastases. It is rational to speculate that the combination of NUCB1 and COX-2 induces the progression of colorectal cancer. Next, we will continue to improve cell experiments to further demonstrate the expression and function of NUCB1 in colorectal cancer cells, and explore whether the combination of NUCB1 and COX-2 induces bevacizumab resistance.

Supplemental Information

Supplemental Information 1 Raw Data.

Click here for additional data file.

Additional Information and Declarations

Competing Interests

Author Contributions

Human Ethics

Data Availability

The authors declare that they have no competing interests.

Yuanyuan Zhang conceived and designed the experiments, analyzed the data, prepared figures and/or tables, and approved the final draft.

Gai Zhang conceived and designed the experiments, analyzed the data, prepared figures and/or tables, and approved the final draft.

Jinghua Zhong conceived and designed the experiments, prepared figures and/or tables, and approved the final draft.

An Li conceived and designed the experiments, performed the experiments, authored or reviewed drafts of the article, and approved the final draft.

Yanyang Wu performed the experiments, authored or reviewed drafts of the article, and approved the final draft.

Zhenli Guo performed the experiments, authored or reviewed drafts of the article, and approved the final draft.

The following information was supplied relating to ethical approvals (i.e., approving body and any reference numbers):

The Scientific Research Logic Committee of the First Affiliated Hospital of Gannan Medical College approved the study (zzsl 2023-104).

The following information was supplied regarding data availability:

The raw data is available in the Supplemental File.

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
