# Peer review of "Expression and correlation of COX-2 and NUCB1 in colorectal adenocarcinoma"

_PeerJ, doi:10.7717/peerj.15774_

## Round 0.1 · original submission · Minor Revisions

Your paper has been carefully reviewed by three reviewers, all of whom provided constructive comments. Although one of them suggested major changes, their concerns can be addressed through adequate revision and discussion. Please revise your paper in light of these comments.

Reviewer 1 ·

Basic reporting

The paper entitled "Expression and Correlation of COX-2 and NUCB1 in Colorectal Adenocarcinoma" by Zhang et al. adopts the method of bioinformatics combined with experimental verification to illustrate the expression and correlation of the two molecules.
The paper could benefit from the following improvements:
In the abstract, the presentation of the results needs to be corrected.
In the introduction, "By inducing cancer cells to secrete…", there is something abrupt about this sentence. Some connecting statements should be added, such as the effect of COX-2 on tumors.
In the introduction, please rewrite the phrase "while reducing the sensitivity of cancer cells to bevacizumab, leading to the emergence of drug resistance."
In the introduction, "is characterized by…", and "which give it many functions", the tense should use the general past tense.
In the results section of the text, the general past tense should be used.
Semi-quantitative integration for the evaluation of immunohistochemistry results lacks references.
Tables 1 and 4 replace "invasion of serosa" with "serosa infiltration".
Figure 2: The picture is not clear; please provide a clear image.
The P value in the text should be written with spaces. It's P< 0.05, not P<0.05.
In the discussion, please rewrite the phrase "COX-2 is an inducible protein, which is nearly not present in normal tissues but can be induced by growth factors…".

Experimental design

In the introduction, have preliminary clinical data been published?
There were only 56 cases in this issue. Can it be expanded?
In Figure 5, the experimental method should be described.
If possible, I hope the authors can conduct cell function experiments to verify in the next step.

Validity of the findings

No comment

Additional comments

To enhance readability, the language may be further optimized.

Reviewer 2 ·

Basic reporting

Thank you very much for inviting me to review the manuscript titled "Expression and Correlation of COX-2 and NUCB1 in Colorectal Adenocarcinoma" (ID: #86134). The authors predicted and verified the expression and correlation of COX-2 and NUCB-1 in colorectal cancer using bioinformatics and immunohistochemistry methods. It has been shown that there is a significant positive correlation between COX-2 and NUCB-1 in colorectal cancer, and both are correlated with several clinical and pathological parameters, indicating that the two may play an important role in the occurrence and development of colorectal cancer. The research methods and ideas are appropriate, but perhaps some improvements can be made in the following aspects:
1.The general past tense needs to be used in the text's outcomes section.
2.There were not enough references for semi-quantitative integration for reviewing immunohistochemistry results.
3.Table 1. “Left semicolon and right semicolon” does not conform to the word specification and can be written as left colon and right colon.
Figure 2 Please offer a clearer image, as the one provided is unclear.
4.It is recommended to indicate the image magnification in Figure 5 of the results section.
5.In the passage above in Figure 3, please clarify the meaning of the “periodic” and rewrite the sentence “However, the PPS time of NUCB1 was periodic without significance.”
6. Figure 4. please identify the proteins corresponding to the plot individually

Experimental design

1.In this edition, there were only 56 cases. Can it be made larger?
2.How many steps were used in the immunohistochemistry experiment in the method section? Two-step method or three-step method? Please explain.
3.Have preliminary clinical data been published in the introduction?

Validity of the findings

None

Additional comments

The language may be further improved to improve readability.

Reviewer 3 ·

Basic reporting

The manuscript mainly elucidated the expression and correlation of COX-2 and NUCB1 in colorectal cancer. There are some issues in the manuscript that need to be addressed.
1. Some of the statements in the manuscript have grammatical and tense errors, and polishing should be performed.
2. Some terms should be used consistently, such as NUCB1 or NUCB-1. The statistical P value should be written correctly as P < 0.05.
3. Figure 2, the picture has some blur and provides a definition of 300dpi or more.
4. In the introduction, “Nucleobindin1 (NUCB1) was first identified in systemic lupus erythematosus (SLE) and is characterized by a variety…”, the tenses should be the same.
5. Figure2. tense errors occur in the description and the past tense should be applied.
6. In the discussion, please replace “Although the sample is relatively small, …” with “Although so few advanced patients were collected, …”

Experimental design

1. Experiment described in the technique section? Method in two steps or in three steps? Please elaborate.
2. There were just 56 examples in this edition. Can you make it bigger in the future?

Validity of the findings

1. In the abstract, the conclusion is too subjective to be drawn from this manuscript. Please reconfirm and rewrite.
2. In the discussion, re-clarify this sentence “We found that there were few relevant studies, such as those on expression level and function, let alone the correlation with COX-2.”, because the introduction section describes some facts about the structural and functional correlation of the two molecules.

Additional comments

None.

---

## Round 0.2 · accepted · Accept

The concerns of all reviewers have been adequately addressed. I believe this research has been improved and strengthened. Their results support the conclusion. This revised paper is ready to be accepted.